# DOG1 as an Immunohistochemical Marker of Acinic Cell Carcinoma: A Systematic Review and Meta-Analysis

**DOI:** 10.3390/ijms23179711

**Published:** 2022-08-26

**Authors:** Vincenzo Fiorentino, Patrizia Straccia, Pietro Tralongo, Teresa Musarra, Francesco Pierconti, Maurizio Martini, Guido Fadda, Esther Diana Rossi, Luigi Maria Larocca

**Affiliations:** 1Division of Anatomic Pathology and Histology, Fondazione Policlinico Universitario A. Gemelli—IRCCS, Largo Agostino Gemelli, 00168 Rome, Italy; 2Division of Anatomic Pathology and Histology, Università degli Studi di Messina—Piazza Pugliatti, 98122 Messina, Italy; 3Unicamillus, Saint Camillus International University of Health Sciences, International Medical University in Rome, 00131 Rome, Italy

**Keywords:** acinic cell carcinoma, immunomarkers, DOG1, salivary gland

## Abstract

DOG1 is a transmembrane protein originally discovered on gastrointestinal stromal tumors and works as a calcium-activated chloride channel protein. There are a limited number of articles on the potential utility of this antibody in the diagnosis of salivary gland tumors in routine practice. In this study, we aimed to investigate the role of DOG1 as an immunohistochemical marker in patients with salivary acinic cell carcinoma (ACC) through meta-analysis. A literature search was performed of the PubMed, Scopus, and Web of Science databases for English-language studies published from January 2010 to September 2021. The literature search revealed 148 articles, of which 20 were included in the study. The overall rate of DOG1 expression in salivary acinic cell carcinoma was 55% (95% CI = 0.43–0.58). Although ACC is a challenging diagnosis, paying careful attention to the cytomorphological features in conjunction with DOG1 immunostaining can help to reach an accurate diagnosis.

## 1. Introduction

Acinic cell carcinoma (ACC) is a low-grade malignant salivary neoplasm. The parotid gland is the predominant site of origin and the median age at diagnosis is 52 years. Possible causes of ACC include previous radiation exposure and familial predisposition. In most cases, this neoplasm has an indolent clinical behavior, and in a minority of cases aggressive behavior; the recurrences and metastases can be seen particularly in the lung and cervical lymph nodes [1]. ACC is histologically defined by serous acinar cell differentiation. However, several cell types and histomorphological growth patterns have been recognized. These include acinar, intercalated ductal, vacuolated, clear-cell, nonspecific glandular, solid-lobular, microcystic, papillary-cystic and follicular growth patterns [2]. The diagnosis of ACCs is frequently difficult, owing to its radiological [3,4] and cytological similarity to benign tumors and to the normal acinar component of the salivary glands, respectively. The differential diagnosis includes, fundamentally, clear cell carcinoma, mucoepidermoid carcinoma, Warthin’s tumor and oncocytoma [5,6]. Both cytomorphologic findings and immunohistochemistry have limited value for discriminating ACCs from salivary gland neoplasms with predominantly oncocytic morphology. Discovered on gastrointestinal stromal tumor protein 1 (DOG1), also known as anoctamin-1/ANO1, is a calcium-activated chloride channel protein made up of eight transmembrane segments that was initially identified in gastrointestinal stromal tumors (GISTs) and is used as an immunohistochemical marker for these neoplasms [7].

The DOG1 role in tumor cell biology is supported by several studies that demonstrated that the overexpression of this protein resulted in an increased tumor cell aggressivity, while its downregulation reduced tumoral cell viability [8,9,10,11,12,13,14,15]. In fact, DOG1 seems to interact with various pathways, such as p38/MAPK [15], EGF/EGFR [13], PI3k/AKT [15,16], TGF/SMAD [14] and many others. Some studies have described a correlation between detectable DOG1 immunostaining and poor prognosis or tumor aggressivity among various cancers of the prostate, breast, ovary, esophagus, stomach, pancreas, head and neck, liver and others [9,10,11,13,14,16,17,18,19,20,21,22,23,24].

However, DOG1 expression has also been detected in normal tissues: in fact, this protein is expressed in pancreatic centroacinar cells, in a subset of islet cells and also in normal acini of salivary glands [17,25]. The explanation of such findings would lie in the fact that, given its properties of the calcium-activated chloride channel, DOG1 protein could have a secretory role in such cell types and studies in murine models have shown that it is essential for the secretory activity of acini in normal salivary glands [26,27,28,29].

On this basis, it is not surprising that DOG1 could be expressed in acinar-derived salivary neoplasms, such as ACC. In fact, a positive DOG1 staining is frequent in ACCs and would support a diagnosis of ACC versus other salivary gland neoplasms. DOG1 positivity can be an admixture of apical membranous, cytoplasmic and complete membranous staining. Interestingly, there are a limited number of studies on the value of DOG1 in salivary glands, particularly focused on ACC [29,30]. Some of these showed different staining patterns, intensity and also an extension of the immunohistochemical reaction. Due to the clonal variability of DOG1 and the limited number of studies, there could be confusion about its role in the diagnosis of ACC. In this study, we aimed to investigate the role of DOG1 as an immunohistochemical marker of salivary ACC through meta-analysis.

## 2. Materials and Methods

A comprehensive literature search in the online databases of Pubmed, Scopus and Web of Science was conducted by searching papers using the keywords “DOG1” and “salivary acinic cell carcinoma” or “ACC” from January 2010 up to September 2021. To try to expand our search, references of the included articles were also screened to identify additional studies. The language was limited to English only.

### 2.1. Study Eligibility

For each included study, the following information was extracted independently in a piloted form: author, country, year of publication, total number of ACC cases, age, sex (% male), tumor size (cm), distant metastases, follow up, outcome and DOG1 immunohistochemical expression in salivary ACC.

### 2.2. Data Extraction

This systematic review was conducted according to PRISMA guidelines (Table 1). Starting from 145 references identified through database searching and 3 additional references identified through other sources, 26 duplicates were removed, and 65 records were excluded with the following reasons: not related to humans (*n* = 24), not related to acinic cell carcinoma (*n* = 10), papers not written in English (*n* = 15), studies published before 2010 (*n* = 16). Some 57 full-text articles were assessed for eligibility and 28 of them were excluded with the following reasons: some studies had no relevant results (*n* = 17), and other criteria (e.g., review article, editorials) (*n* = 11). Finally, 29 studies were included in qualitative synthesis and only 20 references that met the eligibility criteria were retained and included in the current work [29,30,31,32,33,34,35,36,37,38,39,40,41,42,43,44,45,46,47,48].

Data from each eligible study were extracted without modification of original data according to the PICOS: “P”: (population) was constituted by patients with salivary ACC; “I”: (intervention or risk factor) was the ACC group with DOG1 expression, assessed by immunohistochemistry; “C”: “Comparator” was the ACC group without immunohistochemical expression; “S”: “Study design” was the study design of the included studies. Reporting bias across studies was evaluated by a graphic diagnostic tool named funnel plot according to Egger M. et al. [49]. The x-axis in the present analysis is the DOG1 expression and the y-axis is the standard error. In the absence of publication bias, the points representing the studies have a roughly symmetric funnel shape and are distributed about the average effect across the spectrum of levels of precision (Figure 1).

### 2.3. Data Analysis

We aggregated the results of each study using the meta-analytic software ProMeta 2.0 (Internovi, Cesena, Italy). We employed the random-effect model as a conservative approach to account for different sources of variation among studies (i.e., within-study variance and between-study variance) [50]. Q and I^2^ statistics were then conducted to evaluate heterogeneity across studies [51]. Moreover, heterogeneity across study findings was determined using a moderator analysis. Sensitivity analyses were also performed to determine the stability of the study results, computing how the overall rates would change by removing one study at a time. Finally, publication bias analyses were established with two tests: the regression method reported by Egger et al. and the Begg and Mazumdar rank correlation test [49,52]. The absence of publication bias is indicated in both tests by nonsignificant results.

## 3. Results

Based on our criteria, 20 articles that were published between 2010 and 2021 were analyzed and are reported in Table 2.

DOG1 primary antibody clones, dilutions and manufacturers’ specifications regarding immunohistochemical analyses performed in the included studies are presented in Table 3.

All the analyses were performed using automatic staining.

In our study cohort, the median patient age was 54 years (range: 42–66 y/o), and the median tumor size was 3 cm. At the time of follow-up, the analyses also revealed a median of 36.3 months.

Furthermore, the percentage of male participants was 37.04% with a predominantly female population. The shapes of the funnel plots did not reveal evidence of obvious asymmetry (Figure 1).

The results indicated that, in a heterogeneous set of 20 studies, the overall rate of DOG1 expression was 55% (95% CI = 0.43–0.58; Q = 3.12; I2 = 0.00) with a *p* value < 0.05. (Table 4). Fifty-five percent is the value measuring the strength of the relationship between two variables (presence and absence of acinar differentiation) and, in our case, there is quite a strong correlation between the expression of DOG1 in salivary neoplasm and its acinar differentiation (i.e., also a correlation with a final diagnosis of ACC).

These results were highly reliable, as indicated by sensitivity and publication bias analyses (Egger test, −2.09; *p* 0.04; Begg and Mazumdar test, −0.19; *p* 0.57). Details of the overall rates were tested through moderator analyses. Table 5 illustrates the cut-off values for DOG1 in the selected studies.

## 4. Discussion

ACC is a salivary gland malignancy of ductal origin, representing up to 17% of salivary gland neoplasms [53].

The presence of serous acinar cells is a consistent feature of salivary ACC and the main diagnostic criterion is the histologic architecture of the neoplasm, distinguished into four typical patterns: solid, microcystic, follicular and papillary-cystic. ACC is a common cause of false-negative interpretation due to similarity with the normal parotid acinar cells and the absence of hallmark features of malignancy such as necrosis, cellular pleomorphism and high mitotic activity [4]. Thus, there is a need to familiarize with the cytological characteristics of ACC and with its differential diagnoses. The key to the accurate cytological diagnosis of ACC lies in the recognition of the neoplastic acinar cells, with numerous bare nuclei in the background and complete absence of ductal epithelial cells [3].

In the present study, we evaluated the IHC staining profile of DOG1 in patients with salivary ACC through a systematic review and meta-analysis. In the present paper, a total of 20 eligible studies with 333 patients were included. In the overall analysis, the observed expression rate for DOG1 was 55%, showing quite a strong correlation between the expression of this marker and a final diagnosis of ACC.

DOG1 (also known as ANO1, TMEM16 A) is a transmembrane anion channel, which mediates Ca^2+^–dependent Cl- secretion in glands and flat epithelia. Structurally, the DOG1 protein consists of eight transmembrane segments and cytosolic N- and C-termini [54].

Strong DOG1-positive staining ruled out most of the benign entities in the oncocytic salivary gland neoplasm group. In contrast to the weak membranous apical-luminal staining in benign salivary gland acini, DOG1 staining in ACC was moderate to strong, diffuse, membranous and often cytoplasmic [30]. A similar pattern of DOG1 staining was also noted by the study of Chênevert et al. [29], while Hemminger et al. observed a pure luminal pattern of staining [45].

In the overall analysis, we observed that DOG1 expression in salivary tissues is mainly localized in salivary acini, where DOG1 shows a variably intense apical membranous staining and progressively decreases in mucinous acini and intercalated ducts, becoming completely absent more proximally [6]. In our analysis, DOG1 showed a heterogenous positivity in ACCs and the distribution of staining and intensity were moderate to high. Regarding subcellular localization, the majority of the studies included in our work showed diffuse granular cytoplasmic staining in addition to apical-luminal staining and complete membranous staining in some foci. Only Chenevert et al. [29] found a slightly different subcellular localization: in fact, while DOG1 was expressed in some ACCs in their series, it was mostly apical luminal with scattered foci of complete membranous and cytoplasmic staining. Therefore, our findings indicate that DOG1 staining is of pivotal importance in the diagnosis of ACC together with routine hematoxylin and eosin staining that in most cases allows orientation of the diagnosis. Typically, ACCs show readily recognizable serous acinar differentiation on a routine hematoxylin and eosin-stained section, but when this cell type is less prominent, several stainings could help in the diagnosis, such as Periodic Acid-Schiff (PAS) in combination with diastase (PAS-D), mucicarmine, iron stain and in some instances anti-amylase immunostain. However, the sensitivity of these stainings for acinar differentiation is very low. Thus, DOG1 staining offers a sensitive and robust marker to support the diagnosis of ACC. The positivity of DOG1, in fact, is essential to establish the acinar nature of a salivary neoplasm and may represent an ‘exaggerated acinar’ phenotype in ACC, different from neoplasms where DOG1 overexpression is due to gene amplification [29] and can be related to an increased tumor aggressiveness [9,10,11,13,14,16,17,18,19,20,21,22,23,24]. Therefore, DOG 1 is a helpful marker in the diagnostic process of ACCs and its strong positivity can support the diagnosis of these neoplasms. Moreover, the negativity of such markers in our meta-analysis is limited to a minority of cases, many of them represented by poorly differentiated neoplasms; poor differentiation, in fact, entails a partial loss of the acinar phenotype with consequent possible reduction or loss of DDOG1 expression. Based on this finding, DOG1 expression, being prevalent in well-differentiated ACCs, could be interpreted as an index of lower aggressiveness of such neoplasms.

In our experience, we diagnosed 22 ACCs between 2005 and 2021; in this cohort, the ratio male:female was 3:19 (male percentage: 13.63%), the mean patient age was 53.3 years (range: 18–95 y/o), and the mean tumor size was 1.85 cm (range: 0.8–4.5 cm). DOG1 immunostaining, available for only five cases, was performed using Clone SP31 (dilution 1:50; Thermo Scientific, Cheshire, UK) and the I-view 2′-diaminobenzamide (DAB) detection kit (Ventana systems, Tucson, AZ, USA) on a Ventana Benchmark XT automated staining system (Ventana Medical Systems, Inc, Tucson, AZ, USA). Of these five cases, n. 2 ACCs resulted negative, while among the positive ones n. 2 showed focal positivity and only one case showed a strong expression of DOG1 (Figure 2B).

Interestingly, both of the two cases that showed DOG1 negativity in our series were poorly differentiated neoplasms, confirming the findings of the studies included in our meta-analysis. However, our series is really small and not representative; studies on larger cohorts are needed to investigate DOG1 expression in poorly differentiated ACCs.

## 5. Conclusions

Although ACC is a challenging diagnostic entity, paying careful attention to the cytomorphological features of the neoplasm in conjunction with DOG1 immunostaining can help to reach an accurate diagnosis.

## Figures and Tables

**Figure 1 ijms-23-09711-f001:**
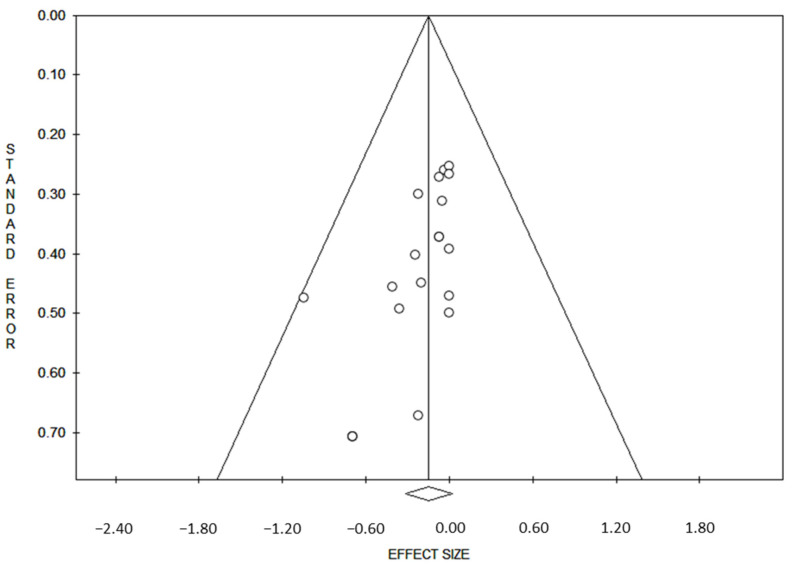
Symmetric funnel plot consistent with lower likelihood of publication bias. The x-axis indicates DOG1 expression and the y-axis is the standard error.

**Figure 2 ijms-23-09711-f002:**
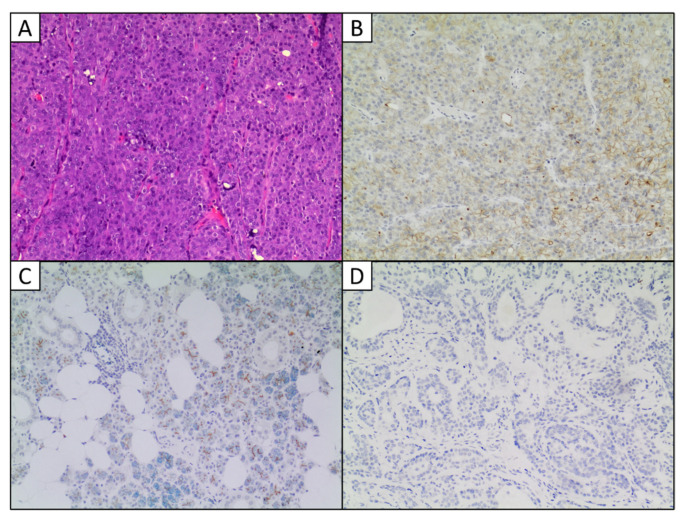
(**A**) Hematoxylin and eosin staining of a case of ACC from our institution (100×). (**B**) Immunohistochemistry showing DOG1 strong membranous expression with cytoplasmic reinforcement in one case of our series of ACCs (100×). (**C**) Immunohistochemistry showing DOG1 apical/luminal staining in normal salivary acini (100×). (**D**) Immunohistochemistry showing DOG1 negativity in a pleomorphic adenoma from our institution (100×).

**Table 1 ijms-23-09711-t001:** Flow diagram of the study selection process.

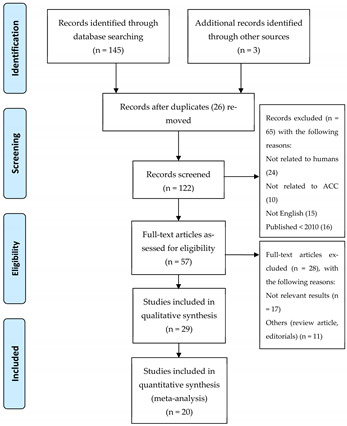

**Table 2 ijms-23-09711-t002:** Characteristics of included studies.

Author	Year	Country	ACC	Age (y/o)	Sex (% M)	Tumor Size (cm)	Distant Metastasis (n. of Cases)	Follow Up (m)	Outcome	DOG1 Expression (%)
Chenevert [29]	2012	USA	28							28/28 (100)
Raboh [31]	2015	Egypt	9							9/9 (100)
Hamamoto [32]	2020	Japan	8	59.8	25	4.1	1	93.4	Alive: 3 Dead: 2 NA: 3	8/8 (100)
Hsieh [33]	2016	Taiwan	28							28/28 (100)
Hsieh [34]	2015	Taiwan	21	42	50	3	12	26		20/21 (95.2)
Khurram [35]	2016	UK	31	46	47.6					31/31 (100)
Khurram [36]	2019	UK	15							14/15 (93.3)
Naous [37]	2017	USA	15	49.3	33.3					14/15 (93.3)
Owosho [38]	2021	USA	6	54	0	2.9				5/6 (83.3)
Said-Al-Naief [39]	2017	USA	14	55	28.5		4	16	Alive 13 Dead 1	11/14 (78.6)
Schmitt [30]	2014	USA	37							32/37 (86.5)
Skaugen [40]	2021	USA	11	66	63.6					9/11 (81.8)
Stevens [41]	2015	USA	13	63	44.4					13/13 (100)
Thompson [42]	2016	USA	25	63.2	36	3.9	22		Alive 6 Dead 16 Lost to FU 3	20/25 (80)
Urano [43]	2014	Japan	6	63	50	2.4	1	10.5	Alive 5 Dead 1	3/6 (50)
Viviane Mariano [44]	2016	Brazil	17	46.2						14/17 (82.4)
Hemminger [45]	2011	USA	5							4/5 (80)
Jung [46]	2013	Korea	6	44.1		2.57		50	Alive 6 Dead 0	3/6 (50)
Shi [47]	2017	China	30							29/30 (96.7)
Kuwabara [48]	2018	Japan	8	55.5	12.5		2			8/8 (100)

Note: ACC: acinic cell carcinoma; NA: not available; FU: follow-up.

**Table 3 ijms-23-09711-t003:** DOG1 primary antibody clones, dilutions and manufacturers’ specifications regarding immunohistochemical analyses performed in the included studies.

Author	Clone	Dilution	Manufacturer
Chenevert [29]	Clone 1.1	1:50	Zeta Co, Sierra Madre, CA
Raboh [31]	Clone 1.1	NA	Thermo scientific
Hamamoto [32]	SP31	RTU	Roche
Hsieh [33]	SP31	RTU	Roche Ventana
Hsieh [34]	SP31	RTU	Roche Ventana
Khurram [35]	NA	1:100	Leica Microsystems
Khurram [36]	Mouse monoclonal	1:250	DAKO
Naous [37]	SP31	RTU	Cell Marque
Owosho [38]	SP31	1:50	Thermo Fisher Scientific
Said-Al-Naief [39]	SP31	RTU	Ventana
Schmitt [30]	SP31	1:40	Cell Marque
Skaugen [40]	SP31	1:50	Thermo Fisher Scientific
Stevens [41]	SP31	RTU	Cell Marque
Thompson [42]	SP31	1:50	Cell Marque
Urano [43]	SP31	1:1	Nichirei
Viviane Mariano [44]	DOG1.1	RTU	Abcam
Hemminger [45]	clone K9	1:100	Leica Microsystems
Jung [46]	Rabbit polyclonal	1:200	Spring Science
Shi [47]	SP31	NA	MXB
Kuwabara [48]	SP31	1:50	Thermo Scientific

Note: NA: not available; RTU: ready to use.

**Table 4 ijms-23-09711-t004:** Summary of meta-analytic results.

	K	N	Overall Rate of Expression (95% CI)	Q	I^2^
**DOG1**	20	333	55% (95% CI = 0.43–0.58)	Q = 3.12	I^2^ = 0.00

Note. K: number of studies; N: number of histological cases available for IHC; CI: confidence interval; I^2^: index for quantifying the degree of heterogeneity; Q: test for heterogeneity; *p* < 0.001.

**Table 5 ijms-23-09711-t005:** Cut-off value for DOG1 in the selected studies.

Author	DOG1 Expression (%)	Cut-Off Value
Chenevert [29]	28/28 (100)	Cases were considered as ‘negative’ if <2% of the tumor expressed DOG1, as ‘focal’ if between 2 and 50%, and as ‘diffuse’ if >50% had staining
Raboh [31]	9/9 (100)	The staining intensity was scored as weak 1+, moderate 2+, and strong 3+. The staining of normal serous acini was used as 2+, more intense staining was graded 3+ and less intense as 1+.
Hamamoto [32]	8/8 (100)	The tumor cells of the ACC cases generally showed strong DOG1 staining intensity on the apical side, but strong cytoplasmic staining was detected in some ACC cases, especially in areas with a solid pattern.
Hsieh [33]	28/28 (100)	Most cases of ACC showed a diffuse (>50% cells) staining, with a mixed apical staining (more frequently observed) and a cytoplasmic staining pattern
Hsieh [34]	20/21 (95.2)	Most cases of ACC showed a diffuse (>50% cells) staining, with a mixed apical staining (more frequently observed) and a cytoplasmic staining pattern
Khurram [35]	31/31 (100)	Strong apical/luminal DOG1 staining was seen in normal acini, although occasional cells demonstrated lateral and basal expression. Staining was stronger in serous acini compared to mucus and focally intercalated ducts showed positive luminal reactivity.
Khurram [36]	14/15 (93.3)	Strong apical/luminal DOG1 staining was seen in normal acini, although occasional cells demonstrated lateral and basal expression. Staining was stronger in serous acini compared to mucus and focally intercalated ducts showed positive luminal reactivity.
Naous [37]	14/15 (93.3)	NA
Owosho [38]	5/6 (83.3)	NA
Said-Al-Naief [39]	11/14 (78.6)	NA
Schmitt [30]	32/37 (86.5)	Immunostaining was graded as weak (1+), moderate (2+), and intense (3+).
Skaugen [40]	9/11 (81.8)	Staining was semiquantitatively scored for intensity (0, 1+, 2+, 3+) and extent (<1%, 1–25%, 26–50%, 51–75%, 76–100%)
Stevens [41]	13/13 (100)	NA
Thompson [42]	20/25 (80)	Luminal DOG1 staining was considered positive and was assessed as a percentage of the respective (LG vs. HG) component being analyzed
Urano [43]	3/6 (50)	NA
Viviane Mariano [44]	14/17 (82.4)	(a) Apical–luminal, (b) mixed membranous and cytoplasmic, (c) cytoplasmic. In the mixed pattern, the membranous component did not exhibit the apical–luminal staining.
Hemminger [45]	4/5 (80)	DOG1 immunopositivity was scored quantitatively for the percentage of positive tumor cells staining (%: 0, ≤10, ≤25, ≤50, >50), intensity (0, negative; 1+, weak staining ⁄ trace, 2+, moderate staining; 3+, strong staining) and subcellular location (cytoplasmic, membranous and luminal).
Jung [46]	3/6 (50)	NA
Shi [47]	29/30 (96.7)	NA
Kuwabara [48]	8/8 (100)	DOG1 was expressed in apical-luminal region

Note. NA: not available.

## Data Availability

The data presented in this study are all available in this manuscript and in the references.

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
