# Peer review of "DOG1 as an Immunohistochemical Marker of Acinic Cell Carcinoma: A Systematic Review and Meta-Analysis"

_ijms, 2022, doi:10.3390/ijms23179711_

Round 1

Reviewer 1 Report

The authors have systemically reviewed the role of DOG1 as immunohistochemical marker of ACC. It may provide useful information in support of the clinical diagnosis of ACC. The manuscript has clear illustration of how the data were extracted, but lack of details of how the data were analyzed. I have some concerns:

11)      Authors stated: “In the present paper, a total of 20 eligible studies with 310 patients were included. In the overall analysis, the observed expression rate for DOG-1was 47%.” A) How does 47% come out? Based on table 4, the smallest DOG-1 Expression (%) is 50. The authors need to show the detailed calculation methods and process.  B). what does 47% mean? The authors need to interpret the significance of 47% in details, not just give a number.

22)      Based on ref. 8 and ref. 15, strong DOG-1 staining was observed in normal acini of salivary gland, that means all ACCs may express DOG-1.  The authors need to give the scientific reasons why they select DOG-1 as an ACC marker, why some AAC express DOG-1, the others do not.

33)      I may not completely agree with “cut-off” criteria in table 4. A Positive ACC case in DOG-1 is a positive ACC case in DOG-1.  For example, ref. 8 stated “All 28 acinic cell carcinomas showed DOG1 staining, and 93% (26/28) tumors showed diffuse (>50%) staining…….. Two acinic cell carcinomas (7%; 2/28) showed only focal DOG1 staining.” Why did the authors classify focal DOG1 staining ACC cases as negative cases? Just because the case number is small? But authors stated from their studies “ ….in those cases, ……while among the positive ones n. 2 showed focal positivity….” .  thus, cut off criteria is not consistent with their statements.     Reference 8 is just an example. The authors need to review all cited documents in detail to readjust the results.  

44)      In Figure 2, authors may need to add proper labels to point out what are Acinar cell components. In figure 3, authors may need to add proper labels to point out which is DOG-1 staining. When I read the manuscript, I have to guess what Acinar cell components are and which is DOG-1 staining. Furthermore, the authors may have to add normal and DOG-1 negative controls to Figure 2 and figure 3 to support their statements in discussion and to convince the readers.

55)      Authors may need to check the whole context of the manuscript to make it be understood clearly. For example, in Abstract, line 16, what is “this antibody”? is it DOG-1 antibody or something else?  “this antibody” is just an example.

66)      Table 2, please correct comma VS decimal point.

77)      Method section lines 66-74, and table 1, please do the math correctly in numbers and includes all references how they were excluded.

Author Response

1
Response to Reviewer 1 Comments
The authors have systemically reviewed the role of DOG1 as immunohistochemical marker of ACC. It may provide useful information in support of the clinical diagnosis of ACC. The manuscript has clear illustration of how the data were extracted, but lack of details of how the data were analyzed. I have some concerns:
11) Authors stated: “In the present paper, a total of 20 eligible studies with 310 patients were included. In the overall analysis, the observed expression rate for DOG-1was 47%.” A) How does 47% come out? Based on table 4, the smallest DOG-1 Expression (%) is 50. The authors need to show the detailed calculation methods and process. B). what does 47% mean? The authors need to interpret the significance of 47% in details, not just give a number.
We have recalculated the % and it was 55%. This is a value measuring the strength of the relationship between two variables (presence and absence of acinar differentiation) and, in our case, there is a quite strong correlation between the expression of DOG-1 in a salivary neoplasm and its acinar differentiation (i.e. also a correlation with a final diagnosis of ACC).
55% is an effect size (ES): In statistics, an effect size is a value measuring the strength of the relationship between two variables (effect and not effect) in a population, or a sample-based estimate of that quantity. It can refer to the value of a statistic calculated from a sample of data, the value of a parameter for a hypothetical population, or to the equation that operationalizes how statistics or parameters lead to the effect size value.
22) Based on ref. 8 and ref. 15, strong DOG-1 staining was observed in normal acini of salivary gland, that means all ACCs may express DOG-1. The authors need to give the scientific reasons why they select DOG-1 as an ACC marker, why some ACC express DOG-1, the others do not.
The positivity of DOG1 is essential to establish the acinar nature of a salivary neoplasm (as demonstrated by the positivity of this marker in normal acinar cells) and may represent an ‘exaggerated acinar’ phenotype in ACC, differently from neoplasms where DOG1 overexpression is due to gene amplification. So, a strong DOG-1 positivity can support the diagnosis of ACC. The negativity of such marker in our meta-analysis is limited to a minority of cases, many of them represented by poorly differentiated neoplasms: poor differentiation, in fact, entails a partial loss of the acinar phenotype with consequent possible reduction or loss of DOG-1expression. This was confermed also by our experience, in fact, in our small series of ACCs, 2/5 cases showed a negative DOG-1 expression, and both cases were poorly differentiated neoplasms. However, our series is really small and studies on larger cohorts are needed to have statistically significant results.
33) I may not completely agree with “cut-off” criteria in table 4. A Positive ACC case in DOG-1 is a positive ACC case in DOG-1. For example, ref. 8 stated “All 28 acinic cell carcinomas showed DOG1 staining, and 93% (26/28) tumors showed diffuse (>50%) staining…….. Two acinic cell carcinomas (7%; 2/28) showed only focal DOG1 staining.” Why did the authors classify focal DOG1 staining ACC cases as negative cases? Just because the case number is small? But authors stated from their studies “ ….in those cases, ……while among the positive ones n. 2 showed focal positivity….” . thus, cut off criteria is not consistent with their statements. Reference 8 is just an example. The authors need to review all cited documents in detail to readjust the results.

Reviewer 2 Report

The paper by Fiorentino et al is a meta-analysis of DOG1 as a immunohistochemical maker in acinic cell carcinoma. The authors selected the studies appropriately, including the potential bias of the studies. Of the 145 articles, the authors selected with their criteria 20, from which they extracted the expression of this marker in ACC, together with expression patterns that may help with an accurate diagnosis. The paper is well-written, and I believe it could be of interest as a biomarker to the biomedical community. 

Author Response

Response to Reviewer 2 Comments
The paper by Fiorentino et al is a meta-analysis of DOG1 as a immunohistochemical maker in acinic cell carcinoma. The authors selected the studies appropriately, including the potential bias of the studies. Of the 145 articles, the authors selected with their criteria 20, from which they extracted the expression of this marker in ACC, together with expression patterns that may help with an accurate diagnosis. The paper is well-written, and I believe it could be of interest as a biomarker to the biomedical community.
We really thank reviewer 2 for the observations made and we hope that the new version of the manuscript could be suitable for publication in IJMS.

Round 2

Reviewer 1 Report

The Authors made a decent, significant revision on the manuscript. they updated figures, tables and interpreted the analyzed results...

I don't have further concerns.